# Validation of a Platform for the Electrostatic Characterization of Textile

Hasan Riaz Tahir *, Benny Malengier , Didier Van Daele and Lieva Van Langenhove

Centre for Textile Science and Engineering, Department of Materials, Textiles and Chemical Engineering, Ghent University, Zwijnaarde, 9052 Ghent, Belgium; Benny.Malengier@UGent.be (B.M.); Didier.VanDaele@UGent.be (D.V.D.); Lieva.VanLangenhove@UGent.be (L.V.L.)
* Correspondence: hasan.tahir@ugent.be

**Abstract:** Floor covering samples of different thickness, pile height, pile design, materials, construction methods, and applied finishes were selected for electrostatic characterization with a standard plotter platform and a newly designed digital platform. There is an existing standard ISO 6356 in which the voltage generated by a human walking on the carpet is measured with human involvement under controlled conditions. A walking person performs the original test procedure to generate the electrostatic charge and manually calculates results. In contrast, the newly designed system does not require a person to calculate peaks and valleys for the generated electrostatic charges, which offers advantages in terms of accuracy, consistency, and reproducibility, and eliminates human error. The electronic platform is extended with an automated foot for a fully automated test, called "automatic mode", that has a fixed capacitive and resistive circuit, in replace of human body resistance, and capacitance that varies from person to person and over time. The procedure includes both the old and new platforms, where the new platform is placed in a "human walking" mode to compare the two and validate the new device. Next, all the floor coverings are tested in automatic mode with the automated foot to compare and validate results. We conclude that the new testing device can fully characterize the electrostatic behavior of textile without the involvement of a human, which offers advantages in terms of accuracy, consistency, and reproducibility.

**Keywords:** electrostatics; floor coverings; automated measuring platform; electrostatic charges

## 1. Introduction

Static electricity may cause an unpleasant, but otherwise harmless, shock when a person touches a door handle after walking just a few steps on a dry, insulated carpet. Static electricity was the first type of electric process known to man [1–3]. This has resulted in the appearance of several excellent, but specialized, treatises on the topic. Nevertheless, there still seem to be numerous misconceptions and misunderstandings about static electricity in textile products [3,4].

Textiles and other materials can be charged with static energy induced by friction. This can be quite a problem, especially with floor coverings, as people walking on them can accumulate high-voltage electrical charge. The discharge of built-up static charge can lead to discomfort for people, influence or damage electronic equipment, or be a fire hazard, for example, at a gasoline station [5]. One of the tests to determine the electrostatic characteristics is the walking test, defined in the ISO 6356 standard [6]. In this test the voltage generated by a person walking on carpet is measured.

Electrostatic charges are produced when two surfaces come in contact with each other and are then separated. When there is contact between the two surfaces, there is a superposition of the atomic fields in the contact area where charges can exchange. If one of the bodies is an insulator, the transferred charges cannot move around, resulting in charge build-up. One of the two bodies will show positive excess of charges while the other

will show a negative excess. Many theories assume that when the material is in contact, charge transfer is only concerned with electrons. Some modern theories also consider two charge exchange mechanisms: electron and positive ion transfer [7]. It is the chemical and physical composition of the materials that mainly determine the polarity and the amount of charge transfer and built-up. Further factors that are important for the charge build-up and transfer of charges between the surfaces are electrical surface resistance and volumetric resistance of the materials. The surface texture, the pressure of contact, the distance of separation, and speed of friction or separation are the main parameters that determine the generation of electrostatic charges. External factors that can affect the generation of electrostatic charges on the surface include the temperature, relative humidity, and the quality and quantity of air around the surfaces. Apart from these main factors, other factors play a role in determining the level of built-up charge on the body. These include footwear, floorcovering, surface coating, walking speed, step height, and step pressure [8–10].

From a body voltage of 6 kV, most people will observe a painful shock. The threshold under which discharge occurs while touching an object is 2kv, while from 4kv in conditions of low relative humidity, while walking on synthetic carpets with an electrically insulated coating, a person may experience a build-up of body voltage up to 25 kV. For laminated floorcoverings, the highest body voltage recorded is 12 kV, equivalent to a discharge energy of 10 mJ [11]. A discharge above 10 mJ can be dangerous for humans. Despite the fact that proof for immediate and adverse effects from a weak electrical field is weak and controversial, it is accepted that having a charged body has an impact on human health [4,10]. In addition to having a painful sensation, spark discharge could lead to dangerous situations such as dropping a heavy or hot flammable liquid, causing injury. This electrical spark can cause ignition of highly flammable materials.

Sensitive electronic circuits could also be at risk of damage from spark discharge. In addition, damage to the computing network in offices or research facilities could cause considerable damage for which a floorcovering manufacturer could be held liable based on the product liability act. A further disadvantage of the build-up of static charge is the attraction of dust, dirt, and smoke particles, resulting in dirty surfaces [2,4,12,13].

Some of the standard test method to evaluate the electrostatic behavior of floorcoverings and laminate are EN 14041, valid since 2004 and amended in 2005 and 2006. It sets the acceptable limits for the categorization of all floor coverings, excluding in flare-up risk regions. The provisions of testing methods talk about EN 1081 for ohm resistance measurement and about EN 1815 and ISO 6356 for measurement of body voltage in a walking test under the controlled condition of 23 °C ± 2 °C and 25% ± 2% relative humidity (RH). The purpose of development of EN 1081 was for resilient floor coverings and the EN 1815 standard for resilient and laminate floor coverings [4,10,14].

Electrostatic charges are produced on a surface by friction or tapping against another surface. There are different modes of electrostatic generation and collection of charges from the tapping and frictional surfaces. If we consider the geometry of electrostatic devices [15], there are four different modes (Figure 1): the vertical contact separation mode [16], the lateral sliding mode [17], the single electrode mode [18], and the free standing sliding mode [19].

Although each geometry of devices based on the two principles, contact electrification and electrostatic induction, there are different parameters like speed of contact, pressure, time of contact, frequency that might have effect on the generation of electrostatic charges [20–22].

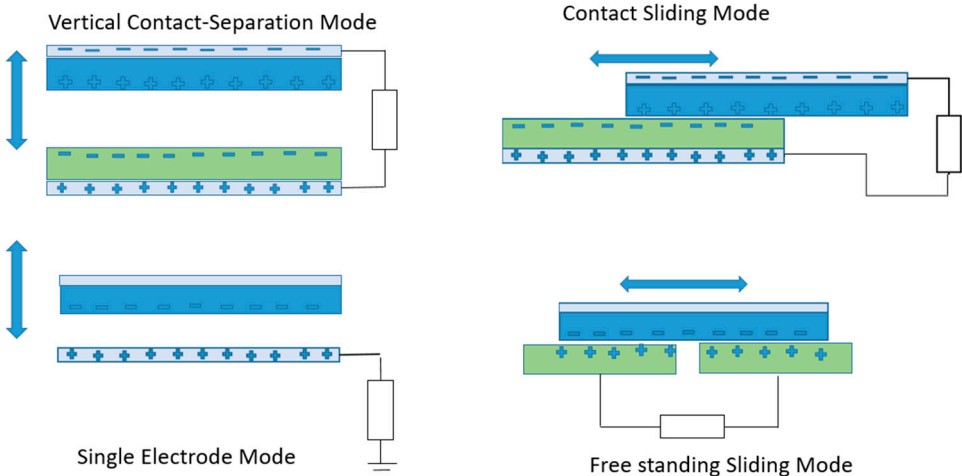

**Figure 1.** Four fundamental modes of electrostatic devices [15].

## 2. Materials and Methods

A total of 10 different floor coverings of different thickness, pile height, pile design, material, construction, and applied finish were selected to compare the new testing device with the standard testing device. The thickness of the floor covering was measured according to the ISO 1765; Machine made textile floorcoverings—Determination of thickness. Basic information about the samples is given in Table 1.

**Table 1.** Basic information about the different floor covering samples, area density (GSM) gram per square meter.

| Sample No. | Area Density (GSM) [g/m²] | Surface Structure | Thickness [mm] | Pile Thickness [mm] | Primary Backing | Secondary Backing |
|---|---|---|---|---|---|---|
| FC-1 | 2339 | Cut Pile | 8.68 | 6.08 | Woven Fabric | PES-Feltback |
| FC-2 | 2583 | Cut Pile | 10.23 | 8.08 | Woven Fabric | |
| FC-3 | 2040 | Cut Pile | 10.17 | 8.19 | Woven Fabric | |
| FC-4 | 2462 | Loop Pile | 9.54 | 5.20 | Non-Woven | PES-Feltback |
| FC-5 | 2651 | Cut Pile | 10.19 | 7.03 | Woven Fabric | |
| FC-6 | 2195 | Cut Pile | 7.68 | 5.41 | Woven Fabric | |
| FC-7 | 1743 | Loop Pile | 8.83 | 6.51 | Woven Fabric | |
| FC-8 | 1862 | Cut Pile | 7.86 | 5.75 | Woven Fabric | |
| FC-9 | 2735 | Cut Pile | 11.04 | 8.53 | Woven Fabric | PES-Feltback |
| FC-10 | 2077 | Cut Pile | 9.60 | 7.20 | Woven Fabric | PES-Feltback |

A new system was designed to measure electrostatic charging. The system is automated in such a way that there was no longer a need for a person to perform the test. The purpose was to provide an accurate, consistent, and reproducible measurement. The steps were performed in a homogeneous pattern across the carpet, resulting in a reliable measurement. The walking commands were based on the protocol for CNC machines, offering the possibility to control the system through a USB connection from a PC. The set-up is shown in Figure 2.

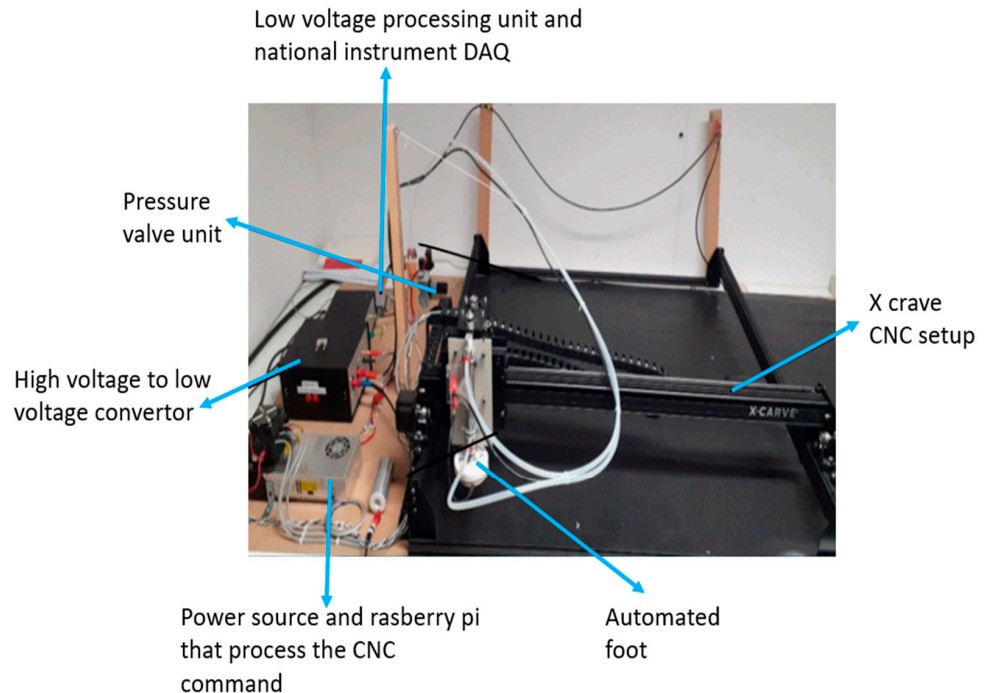

**Figure 2.** Different parts of setup for electrostatic characterization of textiles.

The design provides many parameters that could be varied. The step height could be set to a fixed height, to deliver the same step movement during the entire test. By changing the step height, the influence of this parameter were measured. The regulated pressure for the pneumatic cylinder defined the force applied by the foot on the carpet. Together with the surface area of the sole, it determined the equivalent mass of a person. The stepping frequency was fixed to provide a stable build-up of static charges. The device has an extra testing mode that does not use the automated foot but requires a human to hold the probe and walk on the carpet: this mode is called the human walking mode (HWM). As the first test, the human walking mode was used for the comparison of its results to the already existing standard plotter results (METRAWATT SE 120) making use of a KEITHLY'S Electrometer 610 C, as shown in Figure 3.

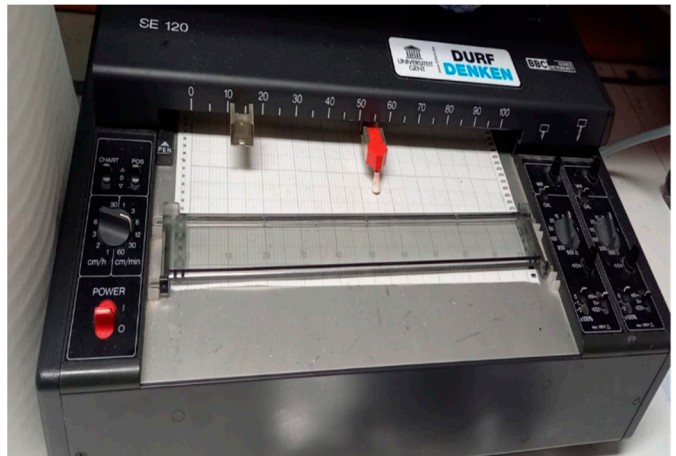
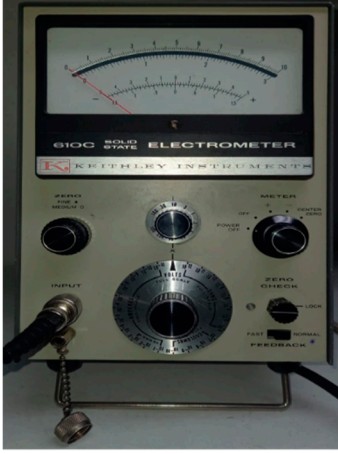

**Figure 3.** Standard plotter (METRAWATT SE 120 with KEITHLY'S Electrometer 610 C to measure and plotting of electrostatic behavior of Floorcoverings.

Figure 4 shows the schematic diagram of the new electrostatic characterization set up. The Human Body Model (HBM) was used to provide an electric equivalent for the test person. This consists of a capacitor of 100 pF and a resistor of 1.5 kΩ. The circuit replaces

the test person in the charging cycle. The generated charge can measure up to 15 kV, which is scaled down by a capacitive voltage divider. The division by 100,000 gives an output of 100 mV when the generated charge is 10 kV, which is an easy to process signal. To avoid problems with the high voltages, all the high voltage components are placed on a separate board made of Perspex. The signal conditioning on the scaled down voltage was done on a normal circuit board. In order not to influence the charge, the input of the circuit was a high impedance buffer. The low bias current high impedance operational amplifier had a neglectable influence on the charge. The input signal was then conditioned further. It was filtered to remove noise at higher frequencies. Two Sallen-Key low pass filters in cascade provided a fourth-order filter with a sharp cut-off at 20 Hz. An extra notch filter at 50 Hz was added in order to reduce the influence of the AC power net as much as possible. The filtered signal was then amplified in order to use the whole range of the ADC. The ADC read the signal at a sampling rate of 100 Hz.

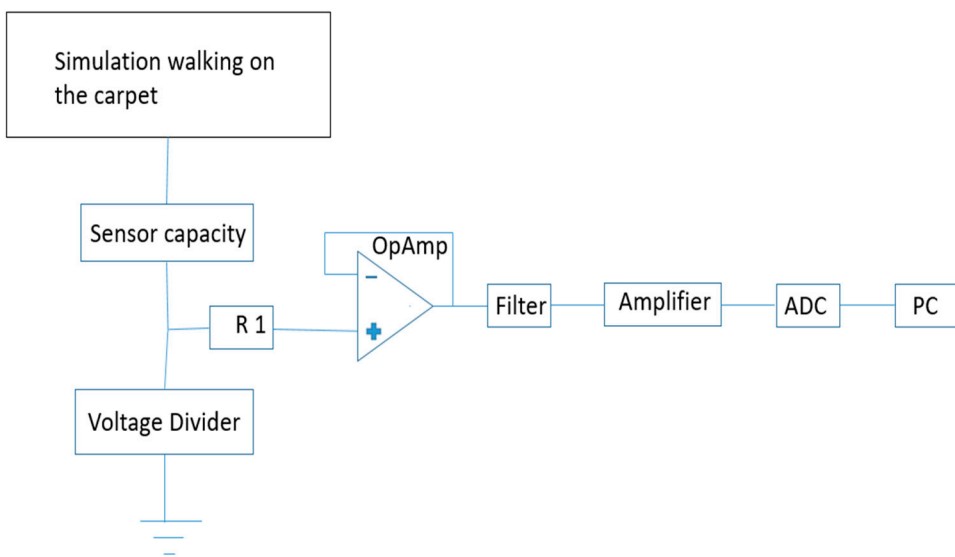

**Figure 4.** Schematic diagram of the new electrostatic characterization set up.

The data read by the ADC were transmitted to a graphical user interface (GUI) on the PC and displayed on a real time chart. It offered direct feedback to the user for optimized control. The user interface also controlled all parts of the mechanical system offering a fully automated test procedure. The GUI thus provided central control over the system, including set-up and calibration of the system. An extra screen was added to provide easy reviewing of the previously recorded data. The goal of the walking test was to determine the maximum accumulated charge over the person or its equivalent. This was done by averaging both the five highest valleys and the five highest peaks of the measured voltage. Once all the data was acquired, an algorithm automatically detected these valleys and peaks. The output was projected on the chart to provide immediate feedback for the user. All results were also saved in a folder of preference, making fast data collection possible.

For the new device, the data was read by a National Instrument DAQ and recorded through a graphical user interface (GUI) on the PC, and displayed on a real-time chart. It offered direct feedback to the user for optimized control. The user interface also controlled all parts of the mechanical system offering a fully automated test procedure. The GUI thus provided central control over the system, including the set-up and calibration of the system. An extra screen was added to provide easy reviewing of the previously recorded data.

The goal of the walking test is to determine the maximum accumulated charge over the person or its equivalent. This is done by averaging both the five highest valleys and the five highest peaks of the measured voltage over a 60 s test interval. Once all the data were acquired, an algorithm automatically detected these valleys and peaks. The output was projected on the chart to provide immediate feedback for the user. All results were also

saved in a folder of preference, making fast data collection possible. The Standard plotter (METRAWATT SE 120) with KEITHLY'S Electrometer 610 C and the probe was used to test the electrostatic charged developed on different floorcoverings according to ISO 6356 as shown in Figure 3 Other necessary tools and auxiliaries are given below

- Test Sandals BAM
- Ethanol (95% conc.) to clean the soles
- Humidity and Temperature sensor to record the environment
- Sample cutting scissor
- Scoured cotton (free of finish or detergent) for cleaning of sole
- Sandpaper (P280 to P360) to clean sandal
- Ionizing Gun to discharge the floorcoverings

The new electronic data acquisition device plotter (DAQ) was custom created at our research group. This device has two modes: human walking mode (HWM), and automated foot mode (AFM). In this paper, the human walking mode is discussed in order to determine if the old plotter, which is the standard testing device, can be replaced by the new automated plotter HWM mode.

### 2.1. Method

Before doing the testing of floorcoverings, conditioning of test samples was performed for 7 days at a temperature of $(23 \pm 2)$ °C and relative humidity of $(25 \pm 2)$% and all tests occurred at these values from standard ISO 6356. Each floor covering sample was evaluated five times on the old plotter and five times with the new DAQ electrostatic evaluating platform in HWM mode. Only one test person was used, and this person was trained to walk according to standard ISO 6356.

The value of generated electrostatic charge for all samples was checked with the standard plotter. The five lowest valleys and five highest peaks in the 60-s testing interval were manually determined according to the set values of different scales. There were three different scales of 10, 30, and 100 on the electrometer that were used for lower, medium, and higher electrostatic value floorcoverings.

The value of the generated electric charge for all samples were next checked with the HWM, where the five lowest valleys and five highest peaks were automatically determined. For the grounding of the floorcoverings an air gun was used. There were five repeats per floorcovering at different positions has been done on each plotter. Then results were compared to check the validity of this test procedure and the new setup. All the floorcoverings were also tested by automated foot of the new device. Figure 5a,b show the air gun set up to remove the residual charges on the floorcoverings. The residual charge was removed before starting each test. Figure 6 shows the graphical user interface for the automated tester with human walking and automatic foot test options. With this GUI (graphical user interface), it is straightforward to change the different testing settings and quickly review the previous testing data.

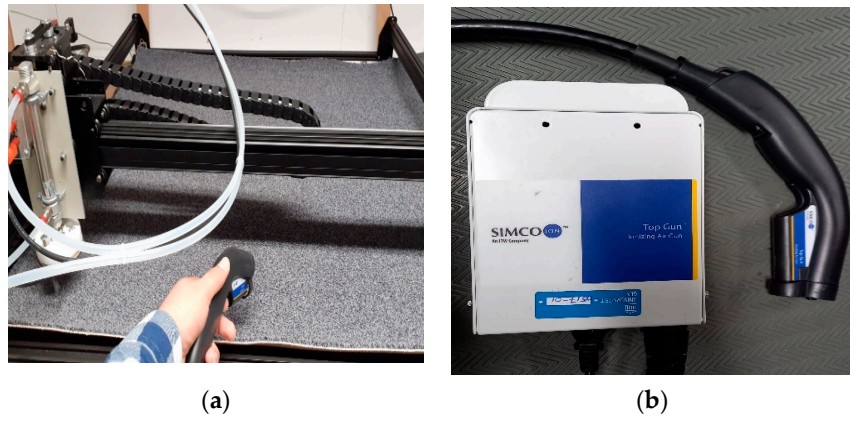

<div align="center">(<b>a</b>)       (<b>b</b>)</div>

**Figure 5.** Setup with air gun to remove the residual charges on the carpet; (**a**) Use of air gun on the floorcovering to remove the residual charge (**b**) air gun.

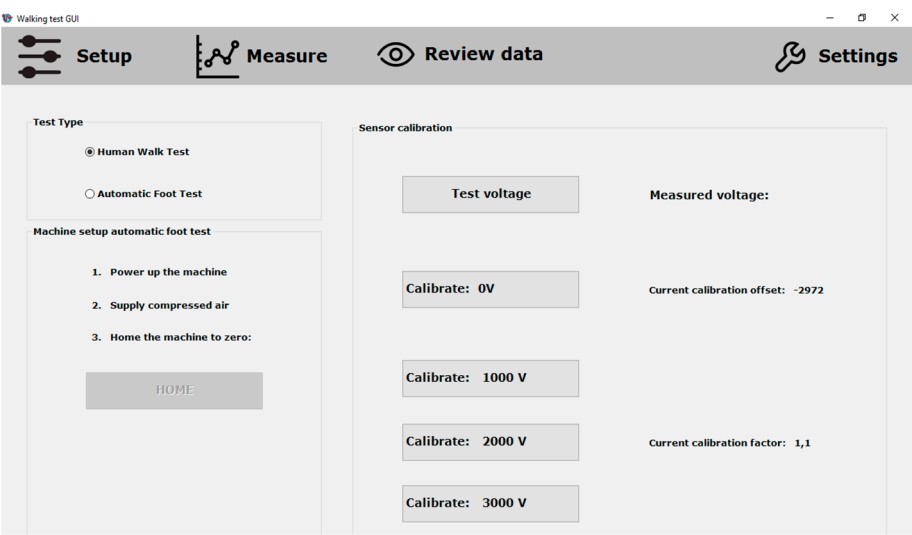

**Figure 6.** Graphical user interface for the automated tester.

*2.2. Statitical Analysis*

The response values for highest peaks and valleys for the standard and new plotter were collected and data was analyzed using Statistical Package for the Social Sciences statistics software (SPSS V.26, IBM Corporation, New York, NY, USA). The responses generated by the 10 floorcoverings on standard and new devices were statistically analyzed including descriptive method and univariate analysis. The level of statistical significance for the analysis was set at $\alpha = 0.05$, and as variables the floorcoverings (sample no given in Table 1) and the platform (old or new), were chosen. To deduce whether the highest peaks and valleys indicate a relationship between the variables (Quality, Platform), the *p*-values were examined. If the *p*-value of HP (Highest Peaks) and HV (Highest Valleys) was greater than 0.05 ($p > 0.05$), then there was not a statistically significant difference for the variables, and vice versa.

*2.3. Effective Graphical Display of Data*

We present bar charts that show the difference of means of HP (Highest Peaks) and HV (Highest Valleys) with standard deviation interval with quality and platform as fixed or X variables.

**3. Results and Discussion**

The electrostatic characterization graph plotted with the standard plotter with the new plotter is shown in Figure 7.

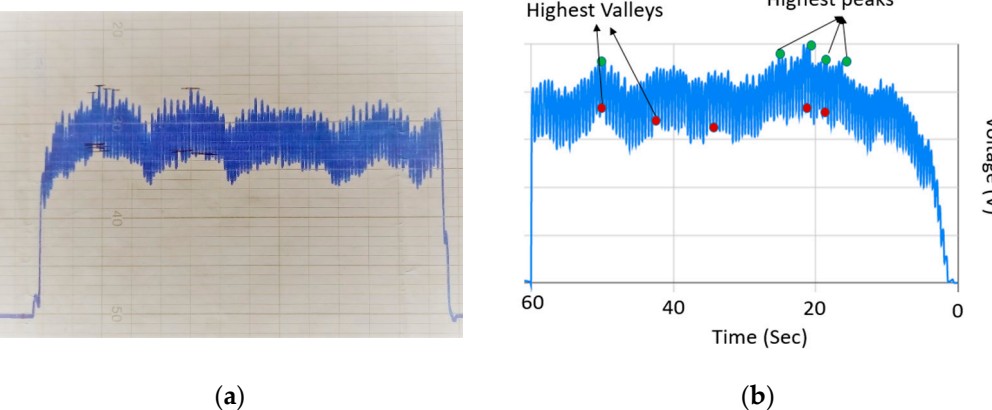

(**a**)  (**b**)

**Figure 7.** Electrostatic characterization graphs (**a**) plotted with standard plotter and manually highlight the five highest peaks and valleys, (**b**) plotted with the new plotter that automatically calculates the five highest peaks and valleys for Human Walking Mode.

### 3.1. Validation of Human Walking Mode of New Device

To validate the human walking mode of the new device, univariate analysis was performed and results are summarized in Table 2, for the highest peaks and highest valleys respectively.

**Table 2.** Univariate analysis results for highest peaks, dependent variable: Highest Peaks (HP).

| Source | Type III Sum of Squares | df | Mean Square | F | Sig. |
|---|---|---|---|---|---|
| Corrected Model | $1.6549 \times 10^9$ [a] | 19 | $8.7098 \times 10^7$ | 3499 | 0.000 |
| Intercept | $1.1473 \times 10^{10}$ | 1 | $1.1473 \times 10^{10}$ | $4.609 \times 10^5$ | 0.000 |
| Floor. Cov | $1.6546 \times 10^9$ | 9 | $1.8384 \times 10^8$ | 7386 | 0.000 |
| Device | 2040 | 1 | 2040 | 0.082 | 0.775 |
| Floor. Cov * Device | $2.581 \times 10^5$ | 9 | $2.868 \times 10^4$ | 1.152 | 0.324 |
| Error | $1.1948 \times 10^7$ | 480 | $2.489 \times 10^4$ | | |
| Total | $1.314 \times 10^{10}$ | 500 | | | |
| Corrected Total | $1.667 \times 10^9$ | 499 | | | |

[a] R Square = 0.993 (Adjusted R Squared = 0.993).

From the univariate analysis as given in Tables 2 and 3, it was found that the highest peaks and highest valleys as determined by the new plotter were not statistically significantly different from the peaks and valleys of the standard device, only the sample type was significant. The graphical representation of the results of all floorcoverings for the highest peaks and valleys are given in Figures 8 and 9 that show the highest peaks (Volt) and highest valleys as dependent variable and with floorcoverings samples and platform (New Device, Std. Device with KEITHLY'S Electrometer 610 C) as variable to show the difference of means and standard deviation as interval. We can conclude that the new plotter can be used as a recording tool for the human walking mode, replacing the old plotter.

**Table 3.** Univariate analysis results for highest valleys, dependent variable: highest valleys (HV).

| Source | Type III Sum of Squares | df | Mean Square | F | Sig. |
|---|---|---|---|---|---|
| Corrected Model | $1.0109 \times 10^9$ [a] | 19 | $5.320 \times 10^7$ | 2441 | 0.000 |
| Intercept | $6.311 \times 10^9$ | 1 | $6.311 \times 10^9$ | $2.896 \times 10^5$ | 0.000 |
| Floor. Cov | $1.011 \times 10^9$ | 9 | $1.123 \times 10^8$ | 5154 | 0.000 |
| Device | 3553 | 1 | 3553 | 0.163 | 0.687 |
| Floor. Cov * Device | $1.282 \times 10^5$ | 9 | $1.424 \times 10^4$ | 0.654 | 0.751 |
| Error | $1.046 \times 10^7$ | 480 | $2.179 \times 10^4$ | | |
| Total | $7.332 \times 10^7$ | 500 | | | |
| Corrected Total | $1.0214 \times 10^9$ | 499 | | | |

[a] R Square = 0.990 (Adjusted R Squared = 0.989).

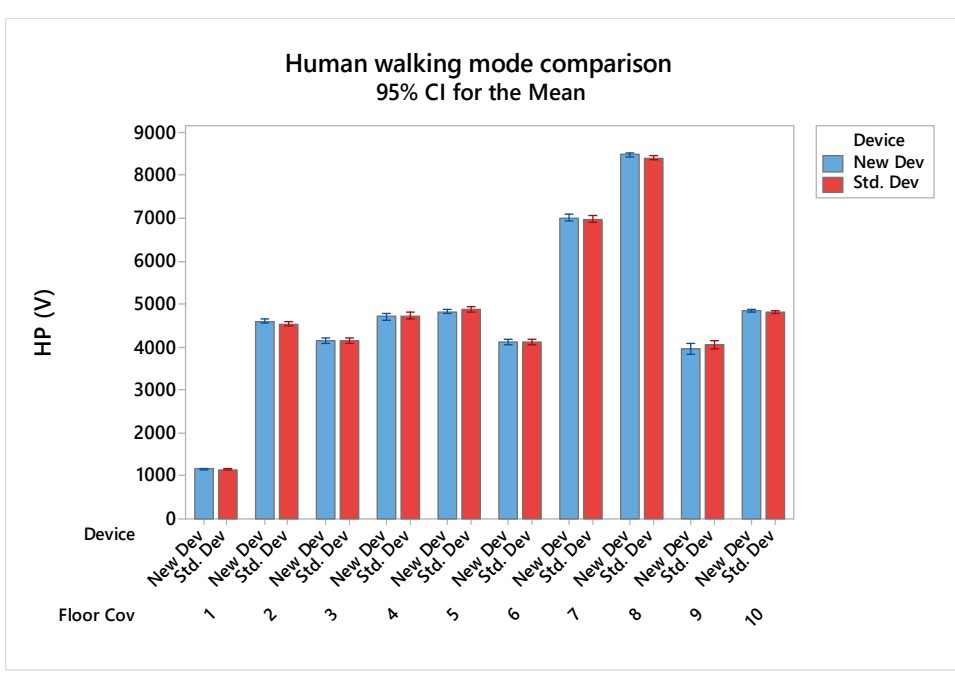

**Figure 8.** Highest peaks (V) with quality of floorcoverings and platform (New, Std.) as X variable to show the difference of means and standard deviation as interval.

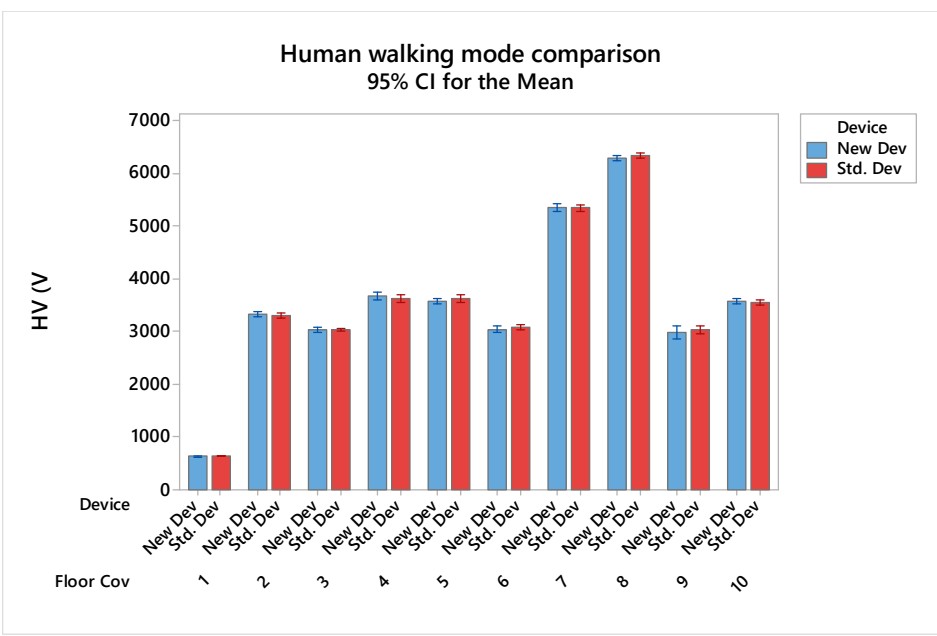

**Figure 9.** Highest valleys (V) with quality of floorcoverings and platform (New, Std.) as X variable to show the difference of means and Standard deviation as interval.

*3.2. Automated Foot for the Charecterization of Floorcoverings*

All the floorcoverings were next characterized with the automated foot mode (AFM) with specific step pattern and with the tapping foot at a specific step height, pressure, and frequency. The frequency of the stepping foot is 120 steps/min, the same as that of used in HWM. The step height is set to 50 mm for the experiments. Pressure is set to 2 bar, with a foot size is 8 cm$^2$, this corresponds to the pressure of a human of 70 kg walking. Figure 10 shows the schematic diagram of the new device with automated foot controlled by G-code, allowing for a specific and controlled movement on the floorcoverings to characterize the electrostatic behavior. Figure 11a,b shows the movement pattern of

automated foot controlled by the G-code to give a controlled motion and electrostatic characterization graphs plotted with the new plotter with the five highest peaks and valleys indicated, respectively.

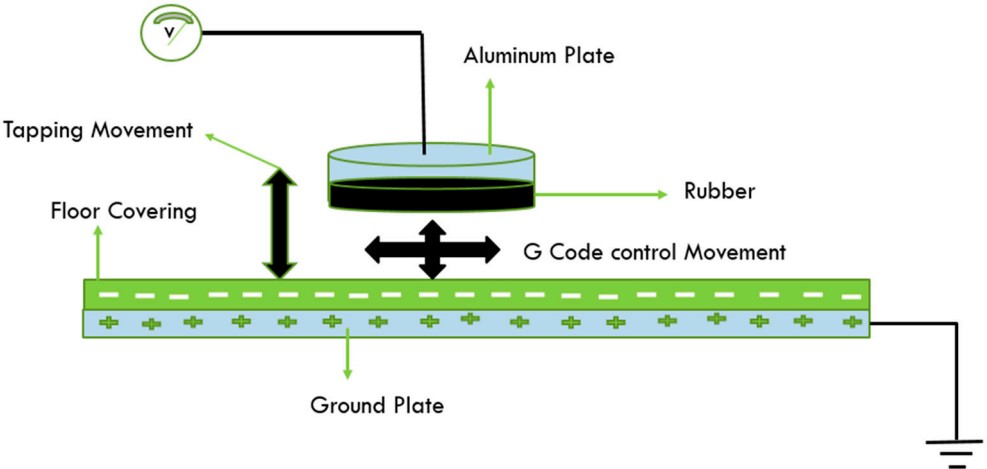

**Figure 10.** Schematic diagram of the new device with automated foot controlled by G-code, allowing for a specific and controlled movement on the floorcoverings for characterize the electrostatic behavior.

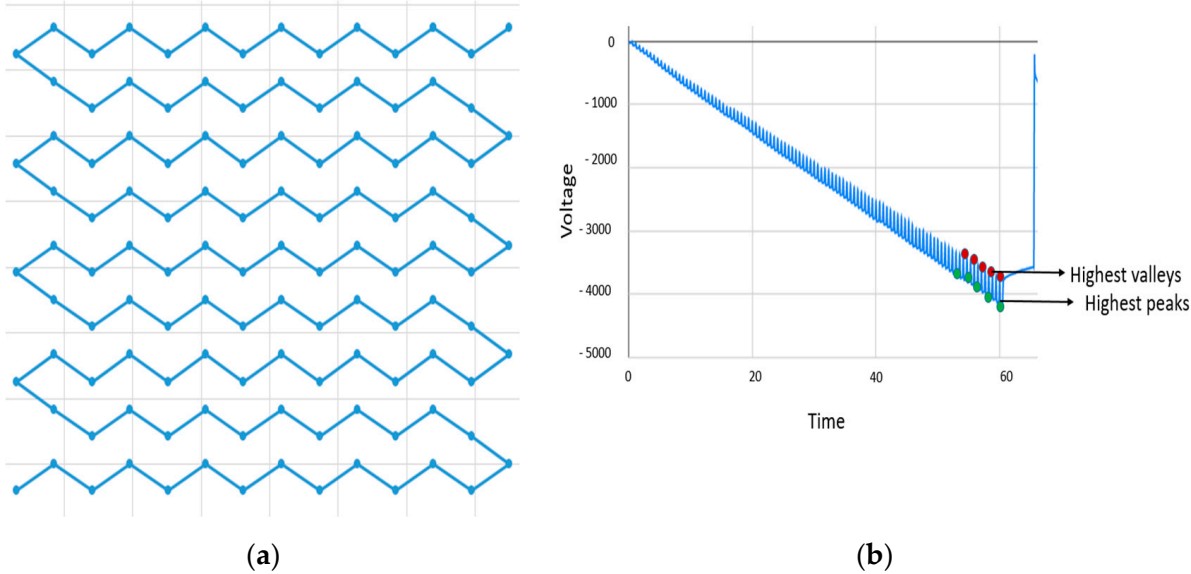

(**a**)                                                    (**b**)

**Figure 11.** (**a**) Movement pattern of automated foot controlled by the G-code to give a controlled motion. (**b**) Electrostatic characterization graphs plotted with the new plotter with the five highest peaks and valleys indicated.

The descriptive statistics of all the floor coverings for the highest peaks and valleys are given in Tables 4 and 5 respectively.

The values of Tables 4 and 5 are graphically depicted in Figures 8 and 9 for the human walking mode of both old and new plotter. We can conclude that human walking mode of the new device could replace the old plotter with advantage of automated calculation and removing the possibility of human calculation error. The AFM consistently has the lowest CV, with sometimes dramatically lower IQR, so is certainly more reproducible than the old plotter and HWM. Ranking the materials from 1 to 10, with 1 the least electrostatic change obtained through the highest peaks, and 10 the highest charge, we obtain the results in Table 6. From this we learn that the ranking of the materials is also different with the AFM.

They are given white (low below 2000 V), light grey (medium 2000–5000 V), and grey (high above 5000 V) color according to the electrostatic voltage generation. The different colors in Table 6. show the different levels of electrostatic charge generation.

**Table 4.** Descriptive statistics of floorcoverings samples for highest peaks checked on old, new, and automated device.

| F. Cov | Device | Count | Mean | SE Mean | St. Dev | CV% | IQR | TR Mean | Min | Max | Range |
|---|---|---|---|---|---|---|---|---|---|---|---|
| | Auto | 25 | 1047 | 5.2 | 26.1 | 2.5 | 35 | 1046 | 1016 | 1104 | 88 |
| 1 | New | 25 | 1158 | 7.146 | 35.729 | 3.1 | 58 | 1157 | 1107 | 1238 | 131 |
| | Old. | 25 | 1148 | 9.928 | 49.641 | 4.3 | 75 | 1147 | 1062 | 1250 | 188 |
| | Auto Foot | 25 | 1951 | 3.819 | 19.093 | 1.0 | 28 | 1951 | 1915 | 1980 | 65 |
| 2 | New | 25 | 4606 | 23.736 | 118.679 | 2.6 | 129 | 4603 | 4456 | 4846 | 390 |
| | Std. | 25 | 4552 | 25.403 | 127.017 | 2.8 | 200 | 4548 | 4380 | 4820 | 440 |
| | Auto Foot | 25 | 1722 | 7.393 | 36.965 | 2.1 | 63 | 1721 | 1657 | 1794 | 137 |
| 3 | New | 25 | 4157 | 25.656 | 128.280 | 3.1 | 194 | 4156 | 3938 | 4399 | 461 |
| | Std. | 25 | 4155 | 27.792 | 138.958 | 3.3 | 220 | 4152 | 3960 | 4420 | 460 |
| | Auto Foot | 25 | 1272 | 5.595 | 27.973 | 2.2 | 39 | 1271 | 1230 | 1329 | 99 |
| 4 | New | 25 | 4727 | 38.723 | 193.617 | 4.1 | 309 | 4727 | 4400 | 5047 | 647 |
| | Std. | 25 | 4733 | 40.396 | 201.980 | 4.3 | 210 | 4725 | 4380 | 5260 | 880 |
| | Auto Foot | 25 | 1933 | 6.093 | 30.465 | 1.6 | 45 | 1934 | 1872 | 1987 | 115 |
| 5 | New | 25 | 4829 | 24.966 | 124.832 | 2.6 | 225 | 4824 | 4694 | 5076 | 382 |
| | Std. | 25 | 4879 | 27.288 | 136.440 | 2.8 | 250 | 4881 | 4640 | 5080 | 440 |
| | Auto Foot | 25 | 1027 | 6.252 | 31.258 | 3.0 | 52 | 1026 | 978 | 1085 | 107 |
| 6 | New | 25 | 4120 | 33.855 | 169.273 | 4.1 | 265 | 4115 | 3914 | 4459 | 545 |
| | Std. | 25 | 4117 | 29.770 | 148.849 | 3.6 | 240 | 4113 | 3920 | 4400 | 480 |
| | Auto Foot | 25 | 2150 | 9.146 | 45.731 | 2.1 | 65 | 2151 | 2054 | 2230 | 176 |
| 7 | New | 25 | 7021 | 38.494 | 192.472 | 2.7 | 394 | 7022 | 6740 | 7260 | 520 |
| | Std. | 25 | 6987 | 39.684 | 198.421 | 2.8 | 410 | 6986 | 6700 | 7300 | 600 |
| | Auto Foot | 25 | 4165 | 10.145 | 50.725 | 1.2 | 72 | 4165 | 4065 | 4273 | 208 |
| 8 | New | 25 | 8493 | 23.992 | 119.961 | 1.4 | 156 | 8496 | 8234 | 8685 | 451 |
| | Std. | 25 | 8424 | 25.994 | 129.971 | 1.5 | 160 | 8430 | 8100 | 8620 | 520 |
| | Auto Foot | 25 | 1377 | 8.382 | 41.909 | 3.0 | 42 | 1375 | 1324 | 1489 | 165 |
| 9 | New | 25 | 3969 | 64.246 | 321.231 | 8.1 | 585 | 3966 | 3460 | 4548 | 1088 |
| | Std. | 25 | 4064 | 40.678 | 203.388 | 5.0 | 240 | 4071 | 3560 | 4400 | 840 |
| | Auto Foot | 25 | 2141 | 6.021 | 30.106 | 1.4 | 38 | 2142 | 2074 | 2189 | 115 |
| 10 | New | 25 | 4843 | 15.643 | 78.214 | 1.6 | 112 | 4839 | 4729 | 5029 | 300 |
| | Std. | 25 | 4824 | 16.052 | 80.260 | 1.7 | 120 | 4820 | 4710 | 5020 | 310 |

The Tables 4 and 5 show the descriptive statistics of floorcoverings samples for the highest peaks and valleys checked on the old, new, and automated devices. It gives a detailed overview of data and statistical parameters, especially coefficient of variation (CV), to show the repeatability and reproducibility under different device modes (human walking either on an old and new device or walking simulator). It shows less variation while using the automated foot device due to having a constant resistive and capacitive circuit. The human involvement in the testing gives the change of body resistance and capacitance that is a cause of more variation in test results. Figures 12 and 13 show the HP (Highest Peaks) and HV (Highest Valleys) graphs to compare automated foot devices and human walking mode devices. It shows that the automated foot device values also increase with the increase in value of human walking devices. It is clear that human walking corresponds to the results, but that the automated foot gives very different results.

**Table 5.** Descriptive statistics of floorcoverings samples for highest valleys checked on standard, new, and automated foot.

| F. Cov | Device | Count | Mean | SE Mean | St. Dev | CV% | IQR | TR Mean | Min | Max | Range |
|---|---|---|---|---|---|---|---|---|---|---|---|
| | Auto Foot | 25 | 918 | 5.466 | 27.328 | 3.0 | 34 | 917 | 886 | 973 | 87 |
| 1 | New | 25 | 632 | 6.860 | 34.300 | 5.4 | 64 | 631 | 584 | 697 | 113 |
| | Std. | 25 | 640 | 5.205 | 26.026 | 4.1 | 31 | 640 | 587.5 | 687.5 | 100 |
| | Auto Foot | 25 | 1743 | 3.373 | 16.863 | 1.0 | 22 | 1743 | 1714 | 1774 | 60 |
| 2 | New | 25 | 3324 | 26.636 | 133.180 | 4.0 | 130 | 3316 | 3196 | 3628 | 432 |
| | Std. | 25 | 3298 | 26.405 | 132.024 | 4.0 | 75 | 3290 | 3186 | 3600 | 414 |
| | Auto Foot | 25 | 1548 | 6.621 | 33.103 | 2.1 | 61 | 1549 | 1481 | 1601 | 120 |
| 3 | New | 25 | 3023 | 24.305 | 121.524 | 4.0 | 220 | 3021 | 2843 | 3265 | 422 |
| | Std. | 25 | 3036 | 14.283 | 71.414 | 2.4 | 110 | 3037 | 2900 | 3160 | 260 |
| | Auto Foot | 25 | 1161 | 9.995 | 49.974 | 4.3 | 96 | 1161 | 1083 | 1243 | 160 |
| 4 | New | 25 | 3683 | 36.725 | 183.625 | 5.0 | 272 | 3680 | 3419 | 4014 | 595 |
| | Std. | 25 | 3630 | 38.125 | 190.624 | 5.3 | 290 | 3621 | 3400 | 4080 | 680 |
| | Auto Foot | 25 | 1755 | 4.807 | 24.034 | 1.4 | 37 | 1756 | 1698 | 1795 | 97 |
| 5 | New | 25 | 3583 | 24.339 | 121.694 | 3.4 | 190 | 3580 | 3420 | 3818 | 398 |
| | Std. | 25 | 3624 | 32.228 | 161.142 | 4.4 | 290 | 3619 | 3400 | 3960 | 560 |
| | Auto Foot | 25 | 930 | 5.225 | 26.126 | 2.8 | 36 | 930 | 885 | 984 | 99 |
| 6 | New | 25 | 3044 | 29.310 | 146.550 | 4.8 | 232 | 3039 | 2829 | 3369 | 540 |
| | Std. | 25 | 3078 | 21.270 | 106.352 | 3.5 | 170 | 3076 | 2900 | 3300 | 400 |
| | Auto Foot | 25 | 1948 | 7.260 | 36.299 | 1.9 | 50 | 1948 | 1869 | 2020 | 151 |
| 7 | New | 25 | 5354 | 34.790 | 173.952 | 3.2 | 345 | 5352 | 5082 | 5675 | 593 |
| | Std. | 25 | 5340 | 34.293 | 171.464 | 3.2 | 290 | 5341 | 5060 | 5600 | 540 |
| | Auto Foot | 25 | 3726 | 11.100 | 55.499 | 1.5 | 86 | 3725 | 3633 | 3840 | 207 |
| 8 | New | 25 | 6295 | 24.870 | 124.350 | 2.0 | 216 | 6296 | 6078 | 6491 | 413 |
| | Std. | 25 | 6327 | 23.622 | 118.110 | 1.9 | 155 | 6330 | 6092 | 6500 | 408 |
| | Auto Foot | 25 | 1247 | 7.540 | 37.699 | 3.0 | 35 | 1246 | 1193 | 1330 | 137 |
| 9 | New | 25 | 2988 | 56.052 | 280.261 | 9.4 | 585 | 2987 | 2612 | 3384 | 772 |
| | Std. | 25 | 3034 | 39.979 | 199.893 | 6.6 | 360 | 3038 | 2600 | 3360 | 760 |
| | Auto Foot | 25 | 1951 | 6.011 | 30.053 | 1.5 | 49 | 1952 | 1887 | 2001 | 114 |
| 10 | New | 25 | 3574 | 23.054 | 115.271 | 3.2 | 180 | 3567 | 3459 | 3835 | 376 |
| | Std. | 25 | 3546 | 24.041 | 120.205 | 3.4 | 189 | 3542 | 3400 | 3790 | 390 |

**Table 6.** Ranking of floorcoverings with respect to the generation of electrostatic charges at different plotters.

| Ranking | Old | New | AFM |
|---|---|---|---|
| 1 | F.Cov. 8 | F.Cov. 8 | F.Cov. 8 |
| 2 | F.Cov. 7 | F.Cov. 7 | F.Cov. 7 |
| 3 | F.Cov. 5 | F.Cov. 5 | F.Cov. 10 |
| 4 | F.Cov. 10 | F.Cov. 10 | F.Cov. 2 |
| 5 | F.Cov. 4 | F.Cov. 4 | F.Cov. 5 |
| 6 | F.Cov. 2 | F.Cov. 2 | F.Cov. 3 |
| 7 | F.Cov. 6 | F.Cov. 6 | F.Cov. 9 |
| 8 | F.Cov. 3 | F.Cov. 3 | F.Cov. 4 |
| 9 | F.Cov. 9 | F.Cov. 9 | F.Cov. 1 |
| 10 | F.Cov. 1 | F.Cov. 1 | F.Cov. 6 |

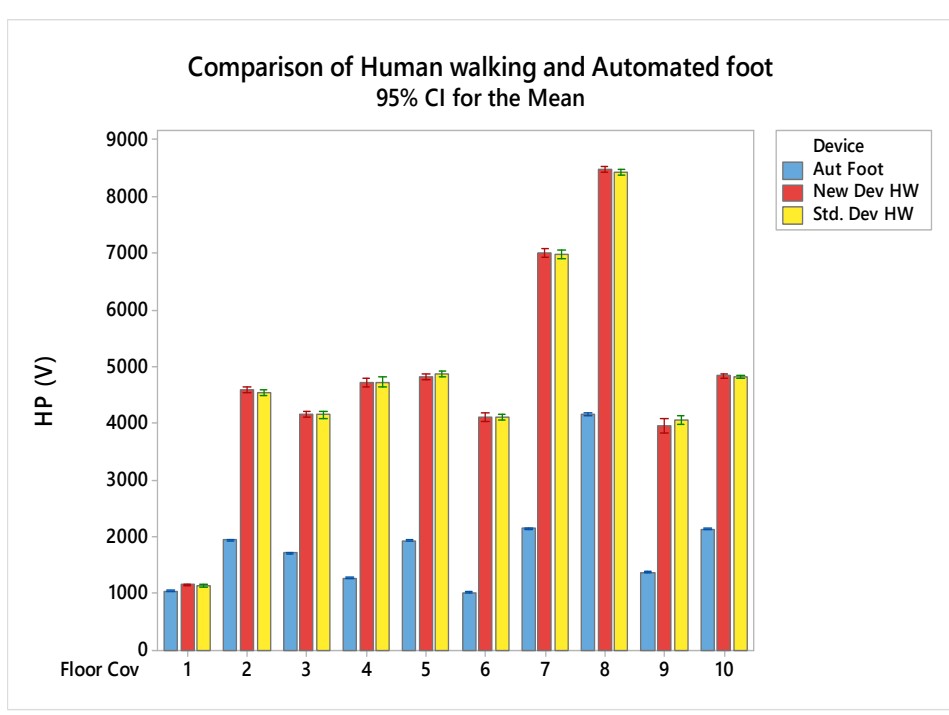

**Figure 12.** HP (Highest Peaks) graph that shows the comparison of automated foot device, and human walking mode devices.

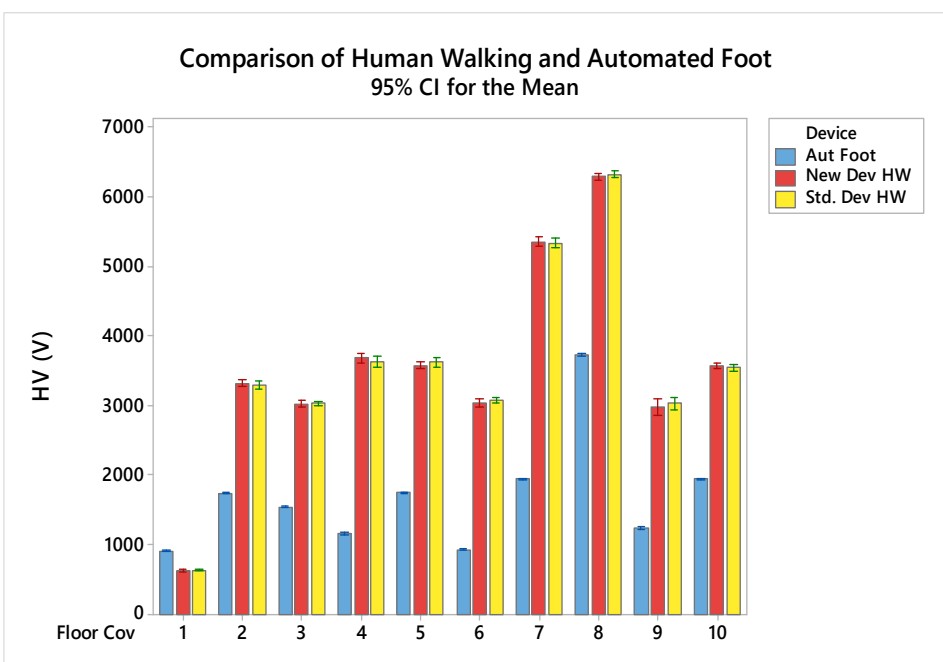

**Figure 13.** HV (Highest Valleys) graph that shows the comparison of automated foot device, and human walking mode devices.

## 4. Discussion and Conclusions

The human walking mode of the new device generates the same results as that of the human walking on the standard device. The automated foot mode with the same frequency of tapping, a specific foot pressure, and height of tapping with 8 cm diameter foot generates a waveform that differs from the human walking on the carpets. The AFM does not correspond but is much more reproducible, making it a better tool for research. However, linking with historical results is not possible, and this should be further researched in the future. The AFM has several parameters that can be optimized and might compare to

HWM, such as the frequency of stepping, the applied pressure, the dwell time and foot downtime of the foot, and the test duration.

Electrostatic charge generation and the resulting electrostatic graph of human walking mode by both devices are the same, validates the new device's human walking mode, and validates that the new custom-made plotter works as well as the old plotter. However, the generation of waveform and electrostatic charges for the automated foot mode is significantly different. More so, the ranking of materials is different. This can be partly attributed to taking the human out of the equation and partly because the parameters of the automated foot have not yet been investigated, and are now selected and fixed to correspond to human walking. Therefore, we can conclude that the AFM is a better tool for research purposes with much lower variability. However, it can, as of now, not be used to compare with historical data obtained with ISO 6356. According to the four fundamental modes of electrostatic devices, the automated foot working principle is based on a single-electrode mode with a variable sample size that has the full ability to characterize the electrostatic behavior of floorcoverings and other textiles in a more repeatable and reproducible way. Though it cannot grasp the specific act of a human walking on a carpet, it can offer an alternative approach to carpet testing.

**Author Contributions:** Conceptualization, B.M., D.V.D., L.V.L. and H.R.T.; methodology, B.M., D.V.D., L.V.L. and H.R.T.; software, B.M., D.V.D. and L.V.L.; validation, B.M., D.V.D., L.V.L. and H.R.T.; formal analysis, B.M., L.V.L. and H.R.T.; investigation, B.M., L.V.L. and H.R.T.; resources, B.M., D.V.D. and L.V.L.; writing—original draft preparation, H.R.T.; writing—review and editing, H.R.T., B.M. and. L.V.L.; supervision, B.M. and. L.V.L.; Project administration, L.V.L. All authors have read and agreed to the published version of the manuscript.

**Funding:** This research was funded by the ICT-Tex project EU project (Nr. 612248-EPP-1-2019-1-BG-EPPKA2-KA) and HEC (Higher Education Commission), Pakistan: HRDI-UESTP Scholarship Project.

**Institutional Review Board Statement:** Not applicable.

**Informed Consent Statement:** Not applicable.

**Data Availability Statement:** Not applicable.

**Conflicts of Interest:** The authors declare no conflict of interest.

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
