# Peer review of "Validation of a Platform for the Electrostatic Characterization of Textile"

_electronics, doi:10.3390/electronics11010115_

Round 1
Reviewer 1 Report
The paper is well written and it is ready for publication with minor corrections. In table 1 please replace GSM with a well defined physical quantity: Area density. In figure 2 black lines are undistinguishable in the picture. I also suggest to uniform the font in the same figure.
Author Response
please see in attachment

Reviewer 2 Report
An engineering approach to instrumentate and quantify manual test methods is extremely important and interesting. On the other hand, an approach that makes the results scientifically easy to understand is also important for creative discussion. Please make the following corrections to the submitted paper.
- You made heavy use of photo and software captures as an overview of your device. You should make a schematic diagram of a unified system block to clear explanation.
- You made heavy use of tables to indicate the test results. However, the key point is not them but the discussion from them. You should try to graph as much as possible.
- You finally discussed the difference between human walking and artificial metal contact devices, but I don't think the conclusion is to bring the contact device closer to human walking. First, you should show a graph that clearly shows the differences between the results of humans and that of machines. It would be even better to show a graph that clearly shows how the correlation differs from human gait in tests with multiple texture changes. And finally, you should show the usefulness of your system.
Round 2
Reviewer 2 Report
All questions are assigned.